# LIGHT (TNFSF14) enhances osteogenesis of human bone marrow-derived mesenchymal stem cells

**Sook-Kyoung Heo[1], Yunsuk Choi[2], Yoo Kyoung Jeong[1], Lan Jeong Ju[1], Ho-Min Yu[1], Do Kyoung Kim[1], Hye Jin Seo[1], Yoo Jin Lee[2], Jaekyung Cheon[2], SuJin Koh[2], Young Joo Min[2], Eui-Kyu Noh[2]\*, Jae-Cheol Jo[1,2]\***

**1** Biomedical Research Center, Ulsan University Hospital, University of Ulsan College of Medicine, Ulsan, Republic of Korea, **2** Department of Hematology and Oncology, Ulsan University Hospital, University of Ulsan College of Medicine, Ulsan, Republic of Korea

\* jcjo97@hanmail.net (JCJ); noheuikyu@gmail.com (EKN)

## Abstract

Osteoporosis is a progressive systemic skeletal disease associated with decreased bone mineral density and deterioration of bone quality, and it affects millions of people worldwide. Currently, it is treated mainly using antiresorptive and osteoanabolic agents. However, these drugs have severe adverse effects. Cell replacement therapy using mesenchymal stem cells (MSCs) could serve as a treatment strategy for osteoporosis in the future. LIGHT (HVEM-L, TNFSF14, or CD258) is a member of the tumor necrosis factor superfamily. However, the effect of recombinant LIGHT (rhLIGHT) on osteogenesis in human bone marrow-derived MSCs (hBM-MSCs) is unknown. Therefore, we monitored the effects of LIGHT on osteogenesis of hBM-MSCs. Lymphotoxin-β receptor (LTβR), which is a LIGHT receptor, was constitutively expressed on the surface of hBM-MSCs. After rhLIGHT treatment, calcium and phosphate deposition in hBM-MSCs, stained by Alizarin red and von Kossa, respectively, significantly increased. We performed quantitative real-time polymerase chain reaction to examine the expressions of osteoprogenitor markers (RUNX2/CBFA1 and collagen I alpha 1) and osteoblast markers (alkaline phosphatase, osterix/Sp7, and osteocalcin) and immunoblotting to assess the underlying biological mechanisms following rhLIGHT treatment. We found that rhLIGHT treatment enhanced von Kossa- and Alizarin red-positive hBM-MSCs and induced the expression of diverse differentiation markers of osteogenesis in a dose-dependent manner. WNT/β-catenin pathway activation strongly mediated rhLIGHT-induced osteogenesis of hBM-MSCs, accelerating the differentiation of hBM-MSCs into osteocytes. In conclusion, the interaction between LIGHT and LTβR enhances osteogenesis of hBM-MSCs. Therefore, LIGHT might play an important role in stem cell therapy.

## Introduction

Osteoporosis is a progressive systemic skeletal disease characterized by bone mineral density reduction and bone quality deterioration that affects millions of people worldwide [1].

**Data Availability Statement:** All relevant data are within the manuscript and its Supporting Information files.

**Funding:** This work was supported by the Basic Science Research Program through the National Research Foundation of Korea (NRF) funded by the Ministry of Education, Science and Technology (NRF-2018R1D1A1B07051343, NRF-2017R1A1A3A04069314). The research was also supported by the Basic Science Research Program through the Biomedical Research Center, funded by the Ulsan University Hospital (UUHBRC-2016-001). The funders had no role in study design, data collection and analysis, decision to publish, or preparation of the manuscript.

**Competing interests:** The authors have declared that no competing interests exist.

**Abbreviations:** TNF, tumor necrosis factor; LTβR, lymphotoxin β receptor; BMCs, bone marrow cells; MSCs, mesenchymal stem cells; BM-MSCs, bone marrow-derived mesenchymal stem cells; HVEM, herpesvirus entry mediator.

Generally, osteoporosis is age-dependent and remains undiagnosed, especially because it remains asymptomatic for several years until a fracture eventually occurs, which consequently limits the daily activities of the elderly [2]. In most people with osteoporotic fractures, the symptoms are non-apparent, yet life-threatening. Osteoporotic fractures can be fatal and are associated with significant high morbidity and mortality rates. Therefore, osteoporosis is considered to be a major health concern and research hotspot globally [1, 3].

Currently, treatments for osteoporosis predominantly rely solely on antiresorptive agents such as bisphosphonates or osteoanabolic agents such as teriparatide [4]. However, these drugs are associated with serious side effects. Therefore, the development of new drugs or treatments is necessary.

Stem cells have immense therapeutic potential, especially in regenerative medicine. In particular, stem cells may be a promising source for cell-based osteoporosis treatments. Therefore, cell-based replacement therapy with mesenchymal stem cells (MSCs) may be an osteoporosis treatment strategy in the future [2, 5]. MSCs are multipotent cells with tremendous potential as a new therapeutic drug [6, 7]. These cells are present in various tissues and organs, including the bone marrow, umbilical cord, adipose tissues, dental pulp, and placenta [8, 9]. They can differentiate into multiple cell types, including those of osteogenic, chondrogenic, and adipogenic lineages [10, 11]. Moreover, MSCs from adult bone marrow can differentiate into various cell types, such as bone, fat, and cartilage tissues, both *in vitro* and *in vivo* [12]. MSCs preferentially move to damaged tissues [7]. Hence, MSCs are primarily used in regenerative medicine as bioreactors of soluble factors that promote tissue regeneration from the progenitors of cells in damaged tissues and possess the ability to differentiate into specific cell types [8]. Furthermore, MSCs can be easily grown *in vitro* and have regenerative properties and potent immunomodulatory activities [12]. MSCs inhibit the functions of dendritic cells (DCs), B cells, and T cells by producing immunomodulatory molecules, such as transforming growth factor beta (TGF-β), liver growth factor, interleukin-10, indolamin 2,3-dioxygenase, and nitric oxide. These properties have made MSCs a promising therapeutic candidate for treating various diseases, including autoimmune diseases [6]. However, there are concerns about the therapeutic approach using MSCs. Because cancer cells and stem cells share several biological properties, transcription factors that monitor the fate of stem cells may play an important role in the regeneration of modified cells in the microenvironment of cancer [13, 14].

LIGHT [tumor necrosis factor (TNF) superfamily protein 14 (TNFSF14), CD258, and lymphotoxin-like inducible protein that competes with glycoprotein D for herpesvirus entry in T-cells] is a cytokine that is primarily expressed in immature DCs, activated T-cells, monocytes, granulocytes, and spleen cells [15]. It has three receptors: herpesvirus entry mediator (HVEM; TNFRSF14, CD270), lymphotoxin β receptor (LTβR; TNFRSF3), and decoy receptor-3 [16]. HVEM is expressed in endothelial cells, DCs, natural killer (NK) cells, T-cells, and B-cells, and LTβR is expressed in fibroblasts and endothelial, epithelial, and stromal cells [17, 18]. Generally, the interaction between LIGHT and HVEM has numerous immunological roles. Their interaction increases cell activation and proliferation as well as cytokine production in T-cells, B-cells, and NK cells [15]. Furthermore, LTβR, which is a membrane-bound receptor of LIGHT, plays an important role in immune system development and immune responses [19]. It also regulates the development and function of the intestinal immune system during embryonic development. In particular, the interaction between LTβR and LTα1β2 initiates the development of lymph nodes and Peyer's patches [20], which is critical for the formation of gut-associated lymphoid tissues.

In our previous study, we showed that LIGHT increases the survival and proliferation of human bone marrow-derived MSCs (hBM-MSCs) via LTβR [10]. However, the role of the interaction between LIGHT and LTβR on osteogenic differentiation in hBM-MSCs is poorly

understood. Therefore, in this study, we investigated the role of LIGHT/LTβR interaction on osteogenesis of hBM-MSCs.

## Materials and methods

### Reagents

Recombinant human LIGHT (rhLIGHT), which was purchased from R&D Systems (Minneapolis, MN, USA), was dissolved in 0.1% bovine serum albumin (BSA)-phosphate buffered saline (PBS) and stored at −20˚C until use. Anti-human CD90-FITC (cat no 555595, 5E10), anti-human CD44-FITC (cat no. 555478, C26), anti-human CD34-FITC (cat no. 348053, 8G12), anti-human CD45-FITC (cat no. 347463, 2D1), anti-LTβR-PE (cat no. 551503, hTNFR-RP-M12), and FITC mouse lgG-isotype control (cat no. 556649, MOPC-21) antibodies were purchased from BD Bioscience (San Diego, CA, USA). Mesenchymal Stem Cell Growth Medium BulletKit™ (MSCGM™, cat no. PT-3001) was obtained from Lonza (Basel, Switzerland). StemPro® Osteogenesis Differentiation Kit and Alizarin red reagent (for staining of osteocytes) were purchased from Invitrogen (Carlsbad, CA, USA). Antibodies such as anti-alkaline phosphatase (ALP, SC-398461), anti-osterix (OSX, SC-393060), and β-actin (SC-47778) for western blotting were purchased from Santa Cruz Biotechnology (Santa Cruz, Dallas, TX, USA). Furthermore, anti-total β-catenin (CST-9562), non-phospho (active) β-Catenin (CST-8814), p-ERK (CST-9106), ERK (CST-4695), p-p38 (CST-4511), p38 (CST-8690), p-JNK (CST-4668), and JNK (CST-9252), anti-rabbit IgG-HRP (CST-7074), and anti-mouse IgG-HRP (CST-7076) antibodies were purchased from Cell Signaling Technology (Danvers, MA, USA). In addition, anti-LTβR antibody (TA332629) was purchased from Origene (Rockville, MD, USA). All reagents were obtained from Sigma-Aldrich (St. Louis, MO, USA).

### BM-derived MSC isolation and culture

Mononuclear cells (MNCs) were isolated from the BM suspension by gradient centrifugation with Lymphoprep (Axis-Shield, Oslo, Norway; density, 1.077 g/mL) and loaded into 100-mm culture dishes containing DMEM (low glucose) with 10% FBS and 1% penicillin and streptomycin. The most common method is based on the capacity of MSCs to adhere to plastic surfaces [21, 22]. After 3-day culture in a humidified incubator at 37˚C and 5% $CO_2$, the non-adhering cells were washed from the flask using PBS. Adherent cells were grown to reach confluence and passaged. After two passages, the cells were cryopreserved in FBS with 10% DMSO. The MSCs used throughout this study were between passage 3 and 8. They were maintained in the MSC basal medium, MSCGM™ in a humidified environment with 5% $CO_2$ at 37˚C.

### Ethics approval

All human-related methods were conducted in accordance with the relevant guidelines and regulations. All patients were provided with written informed consent prior to the commencement of the study, and written informed consent was obtained from all patients. The study protocol and patient consent form and information were approved by the Ulsan University Hospital Ethics Committee and Institutional Review Board (UUH-IRB-2016-07-026).

### Phenotypic analysis using flow cytometry

The cells were harvested and washed twice with fluorescence-activated cell sorting (FACS) buffer (PBS containing 0.3% BSA and 0.1% $NaN_3$) and incubated with various antibodies against each cell surface antigen, such as CD19, CD34, CD45, CD44, CD90, CD105, and

LTβR, on ice for 30 min. Cells were then washed twice with FACS buffer and analyzed using the FACSCalibur flow cytometer and CellQuest Pro software (BD Bioscience).

## Alizarin red and von Kossa staining

On day 18 after osteogenic induction, the deposition of the minerals calcium and phosphate was assessed using Alizarin red and von Kossa staining, respectively. For Alizarin red staining, cells were fixed with 4% paraformaldehyde for 10 min at room temperature and washed three times with distilled water. Then, 2% Alizarin red was added, and the cells were incubated for 15–30 min at room temperature and then rinsed with distilled water. For von Kossa staining, cells were fixed with 4% paraformaldehyde for 10 min at room temperature and washed three times with distilled water. Next, 5% silver nitrate was added, and the cells were exposed to ultraviolet radiation in the dark for 30 min. To eliminate nonspecific staining, 5% sodium thiosulfate was added for 5 min. The samples were then analyzed using a microscope (Olympus, NY, USA).

## Quantitative real-time reverse transcription-polymerase chain reaction (RT-qPCR)

Using TRIzol reagent, we isolated total RNA from cells. Then, cDNA was synthesized using Superscript III reverse transcriptase according to the manufacturer's instructions and analyzed with the CFX96 real-time PCR system using iQ SYBR Green Supermix (Bio-Rad, Hercules, CA, USA). All primers were synthesized by Bioneer Corporation (Daejeon, Korea). The primer sequences are shown in Table 1. The PCR conditions were as follows: an initial denaturation step at 95°C for 3 min and 40 cycles at 95°C for 10 s, 63°C for 10 s, and 72°C for 30 s. During the 72°C extension step, optical detection was performed to determine SYBR green fluorescence. The expression of the target gene relative to that of the endogenous control gene glyceraldehyde-3-phosphate dehydrogenase was calculated using the difference in the threshold cycle method ($\Delta C_t = C_{t'}$ target $- C_{t'}$ control), as described in our previous report [23], in

**Table 1. Sequences of primers used for quantitative RT-PCR.**

| Target gene | Primer sequences (5′→3′) | |
|---|---|---|
| RUNX2 | Forward | CAGATGGGACTGTGGTTACTG |
| | Reverse | AGATCGTTGAACCTTGCTACTT |
| COL1A1 | Forward | CTAAAGGCGAACCTGGTGAT |
| | Reverse | TCCAGGAGCACCAACATTAC |
| ALP | Forward | GGAGTATGAGAGTGACGAGAAAG |
| | Reverse | GAAGTGGGAGTGCTTGTATCT |
| OSX | Forward | GCAAAGCAGGCACAAAGAAG |
| | Reverse | CAGGTGAAAGGAGCCCATTAG |
| OCN | Forward | TCACACTCCTCGCCCTATT |
| | Reverse | CCTCCTGCTTGGACACAAA |
| LTβR | Forward | GGCACCTATGTCTCAGCTAAAT |
| | Reverse | GTAGTTCCAGTGCTCGTTGTAG |
| GAPDH | Forward | GATCATCAGCAATGCCTCCT |
| | Reverse | GTCATGAGTCCTTCCACGATAC |

RUNX2, Runt-related transcription factor 2; COL1A1, Collagen, type I, alpha 1; ALP, Alkaline phosphatase; OSX, Osterix; OCN, Osteocalcin; LTβR, Lymphotoxin beta receptor; GAPDH, Glyceraldehyde 3-phosphate dehydrogenase.

which the relative expression equaled $2^{-\Delta\Delta Ct}$ ($\Delta\Delta C_t = \Delta C_{t'}$ target $- \Delta C_{t'}$ untreated). All assays were performed in triplicate.

## Analysis of ALP activity

To measure ALP activity, cells were lysed in RIPA buffer, and 10 μl of lysate was incubated with 90 μl of fresh solution containing the substrate *p*-nitrophenyl phosphate at 37˚C for 20–30 min. To stop the reaction, we added 100 μl of 0.5 N NaOH. Then, we measured the absorbance at 405 nm on a SpectraMax iD3 Microplate Reader (Molecular Devices, San Jose, CA, USA). We determined the total protein concentration using a BCA protein assay kit (Thermo Fisher Scientific, Waltham, MA, USA). The results are expressed as fold changes from the base conditions (undifferentiated control cells) using four–five culture wells for each experimental condition.

## Western blotting analysis

The samples were washed three times with ice-cold PBS and harvested. Western blotting was performed as previously described [24, 25]. The blots were developed using the ChemiDoc<sup>TM</sup> Touch Imaging System and analyzed using Image Lab<sup>TM</sup> software (Bio-Rad).

## Immunofluorescence

Cells were cultured in an induction medium placed in a 60-mm dish. We detected ALP and total β-catenin using a fluorescence microscope (Olympus, NY, USA). Briefly, cells were fixed with 4% paraformaldehyde for 10 min at room temperature, permeabilized, and then blocked for 30 min in 0.1% Triton X-100 and 5% BSA. Subsequently, fixed cells were washed and incubated for 120 min at room temperature with anti-ALP (1:100; Santa Cruz Biotechnology) and total β-catenin (1:400; Cell Signaling Technology). We incubated the cells with a fluorescence-conjugated secondary antibody (Thermo Fisher Scientific, Waltham, MA, USA) for 60 min at room temperature and stained the nuclei with DAPI (Sigma) for 10 min.

## Statistical analysis

Data are expressed as means ± the standard error of the mean (SEM) based on at least three independent experiments. All values were evaluated using one-way analysis of variance, followed by Tukey's range test using GraphPad Prism version 7.0 (GraphPad Software, Inc., La Jolla, CA, USA). Differences were considered significant when $p < 0.05$. Each assay was conducted in triplicate.

# Results

## LIGHT promotes bone formation in hBM-MSCs

In our previous study, we stained cells undergoing adipogenesis (Oil Red O staining), chondrogenesis (Alcian blue staining), and osteogenesis (Alizarin red staining) to confirm the quality of MSCs [10]. As shown in S1A and S1B Fig, we identified negative and positive differentiation markers of hBM-MSCs using flow cytometry analysis. The MSCs used in our experiments were consistently negative for CD34, CD45, and CD19 and positive for CD90, CD44, and CD105. In addition, LTβR was constitutively expressed on the surface of hBM-MSCs (S1C Fig).

Bones are calcified tissues containing calcium and phosphorus, and they play important roles in supporting and protecting the body. Bone calcification is the process of depositing mineral salts on the collagen fiber matrix, which crystallizes and hardens the tissue. Deposition of calcium was assessed using Alizarin red staining from 7 days up to 21 days after osteogenic

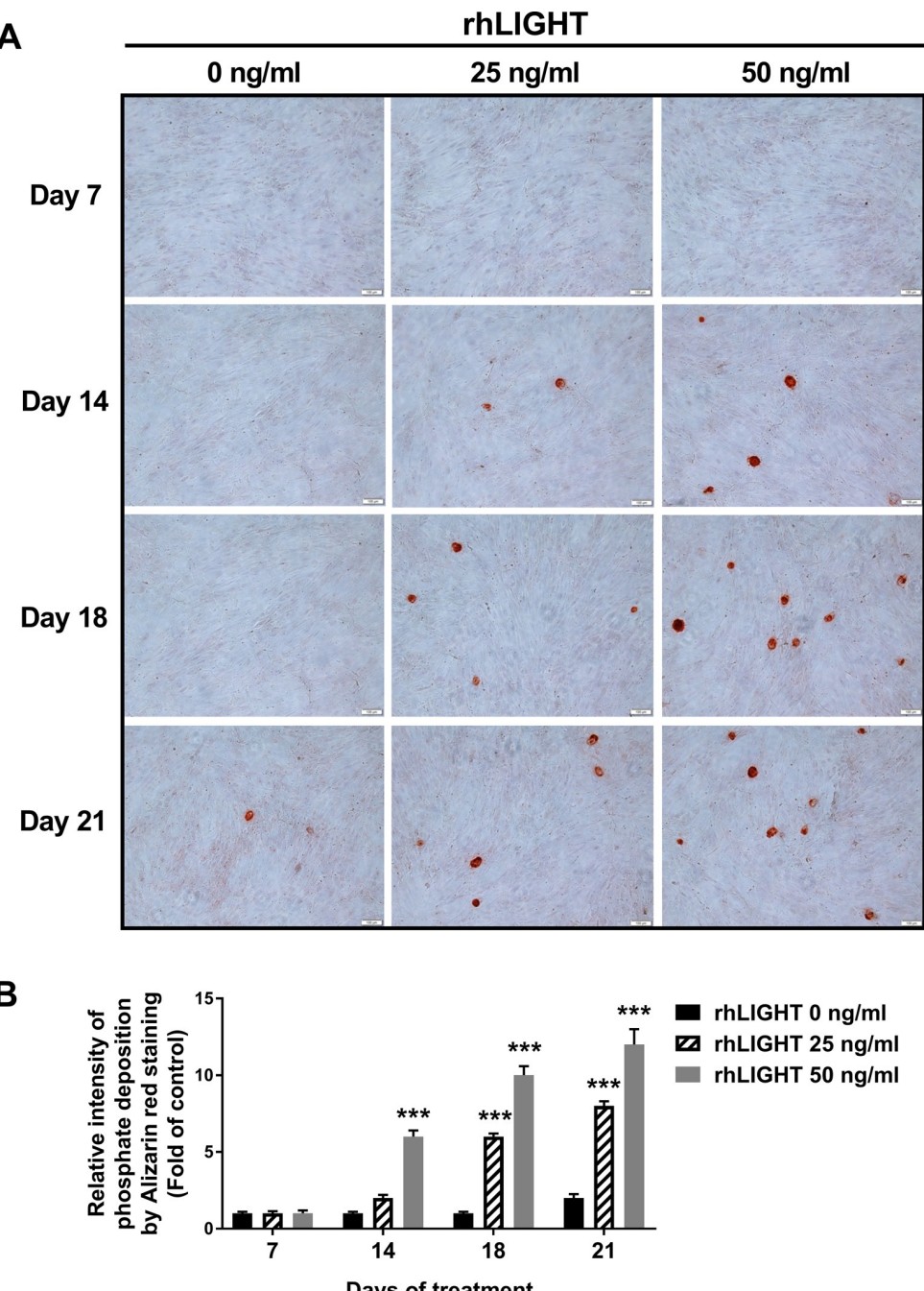

**Fig 1. RhLIGHT increases calcium deposition in hBM-MSCs.** (A) Cells were incubated with 0, 25, and 50 ng/ml concentrations of rhLIGHT for 7, 14, 18 and 21 days and then stained with 2% Alizarin red to confirm calcium deposits, as described in the Materials and Methods. Microscope magnification: ×100, Scale bar = 100 μm. (B) The number of calcium deposition spots in Fig 1A is summarized by Fig 1B. Data are expressed as mean ± SEM. Significantly different from the control cells (*); ***, $P < 0.001$.

induction by rhLIGHT, as shown in as shown in Fig 1. In the samples on day 14 and day 18, calcium deposition was observed only in the rhLIGHT treatment group. Moreover, in the sample on day 21, calcium deposition began to be observed in the control group, and the degree of calcium deposition was accelerated in the rhLIGHT treatment group. Relative intensity of

## A

### rhLIGHT

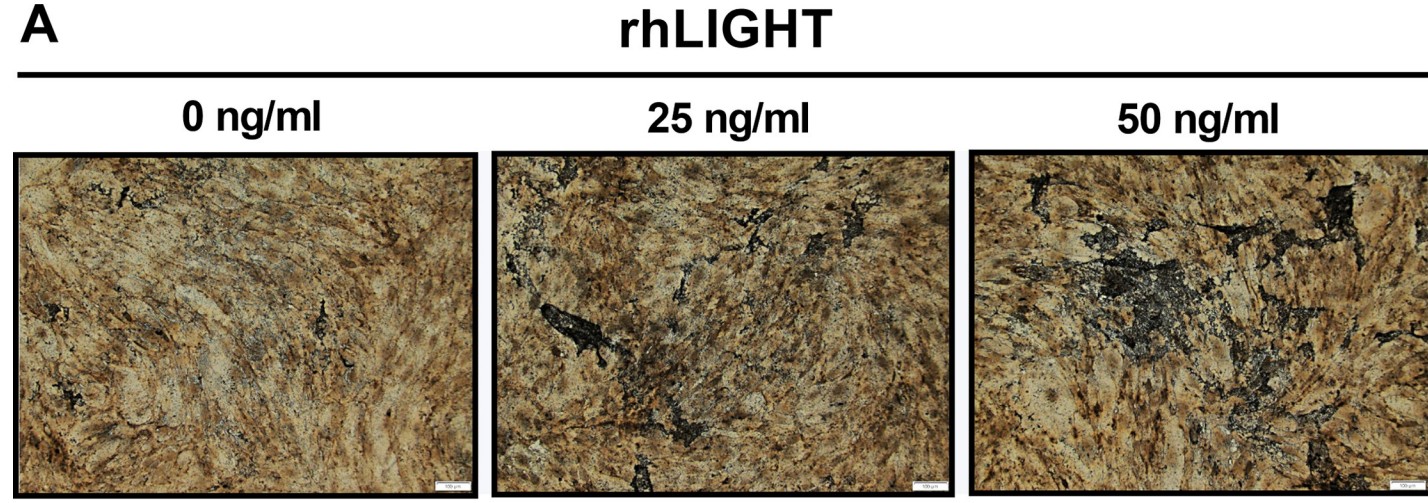

| 0 ng/ml | 25 ng/ml | 50 ng/ml |

## B

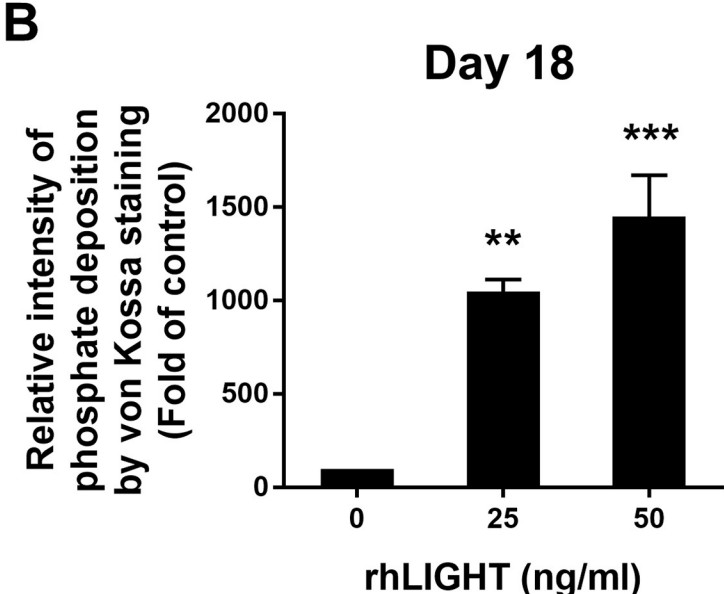

**Fig 2. RhLIGHT increases phosphate deposition in hBM-MSCs.** Cells were incubated with 0, 25, and 50 ng/ml concentrations of rhLIGHT for 18 days and stained by von Kossa to confirm phosphate deposition, as described in the Materials and Methods. Microscope magnification: ×100, Scale bar = 100 μm. (B) Relative intensity of phosphate deposition by von Kossa staining in Fig 2A is summarized by Fig 2B. Data are expressed as mean ± SEM. Significantly different from the control cells (*); **$P < 0.01$; ***$P < 0.001$.

phosphate deposition by Alizarin red staining in Fig 1A is summarized by Fig 1B. According to Fig 1, rhLIGHT effect was better on day 18 than day 21. Accordingly, we used these conditions for the remainder of the experiments. Furthermore, deposition of phosphate was assessed using von Kossa staining, on day 18 after osteogenic induction by rhLIGHT. As a result, rhLIGHT increased phosphate deposition in hBM-MSCs (Fig 2). These results indicate that LIGHT promotes bone formation in hBM-MSCs via its receptor protein, LTβR (S1 Fig, Figs 1 and 2).

## LIGHT enhances the expression of osteoblast differentiation markers and ALP as well as ALP activity in hBM-MSCs

To examine the role of rhLIGHT in osteogenic differentiation, we examined the levels of osteos-specific genes and proteins, including RUNX2, collagen I alpha 1 (COL1A1), ALP, OSX/Sp7, and osteocalcin, using RT-qPCR and western blotting analysis. RT-qPCR analysis revealed that the mRNA levels of ALP, RUNX2, COL1A1, OSX, and OCN on day 18 were significantly high following treatment with 50 ng/ml rhLIGHT treatment (Fig 3A). Western blotting analysis revealed that the levels of ALP, COL1A1 and OSX proteins were higher on day 18 than on day 0 following treatment with 50 ng/ml rhLIGHT (Fig 3B, left panel). On day 18, the expression of ALP and OSX proteins was considerably induced in the 50 ng/ml rhLIGHT treatment group compared with that in the 0 ng/ml rhLIGHT control group (Fig 3B, right panel). We also performed immunofluorescence to confirm the expression of ALP; its expression increased in the 50 ng/ml rhLIGHT treatment group compared with that in the 0 ng/ml rhLIGHT control group on day 18 (Fig 3C, Scale bar = 100 μm). Additionally, we evaluated ALP activity, which is a marker of bone formation, on days 0 and 18 following treatment with 50 ng/ml rhLIGHT, which induced osteogenic differentiation. On day 18, ALP activity was evaluated for each rhLIGHT concentration (0 and 50 ng/ml). In the 50 ng/ml rhLIGHT group, the ALP activity on day 18 was 7.8-fold higher than that on day 0. On day 18, ALP activity in the 0 ng/ml rhLIGHT group was 3.4-fold higher than that in the 50 ng/ml rhLIGHT group (Fig 4D).

## LIGHT activates the WNT/β-catenin and p38 MAPK signaling pathways in hBM-MSCs

It is well-known that Wnt/β-catenin signaling is critical in promoting osteoblastic differentiation and maintaining bone mass [26, 27]. In addition, increased Wnt/β-catenin signaling generally exhibits proosteoblastic and antiadipocytic differentiation effects. To examine the signaling pathways involved in the osteogenic differentiation of hBM-MSCs following treatment with rhLIGHT, we assessed the common signaling pathways involved in osteogenesis using western blotting analysis. These pathways included the MAPK signaling pathway, PI3K/AKT signaling pathway, and Wnt/β-catenin pathway. On day 18 of osteogenic differentiation, total β-catenin and activated β-catenin expression levels were high in the rhLIGHT group (Fig 4A, Left panel). Specifically, compared with those in the 0 ng/mL rhLIGHT group, the expression levels of total β-catenin and activated β-catenin were significantly induced in the 50 ng/ml rhLIGHT group (Fig 4A, left panel). In addition, the expression level of p-p38 MAPK was slightly increased following rhLIGHT treatment in a dose-dependent manner (Fig 4A, right panel). No significant differences were found in the other MAPK signaling pathways (Fig 4A) or the PI3K/AKT signaling pathway between these groups (data not shown). We also performed immunofluorescence to confirm the expression of total β-catenin, which was found to be increased in the 50 ng/ml rhLIGHT treatment group compared with that in the 0 ng/ml rhLIGHT control group on day 18 (Fig 4B, Scale bar = 100 μm). Therefore, rhLIGHT-induced osteogenesis of hBM-MSCs was strongly mediated by WNT/β-catenin pathway activation, which accelerated the hBM-MSCs to differentiate into osteocytes. Thus, LIGHT/LTβR interaction increases osteogenesis of hBM-MSCs (Fig 4).

## LIGHT enhances the expression of its receptor LTβR in hBM-MSCs

We examined the effects of rhLIGHT on the expression of its receptor, LTβR, in hBM-MSCs. We confirmed that membrane LTβR was constitutively expressed on hBM-MSCs (S1C Fig). In addition, the LTβR expression level changed depending on the concentration of rhLIGHT in

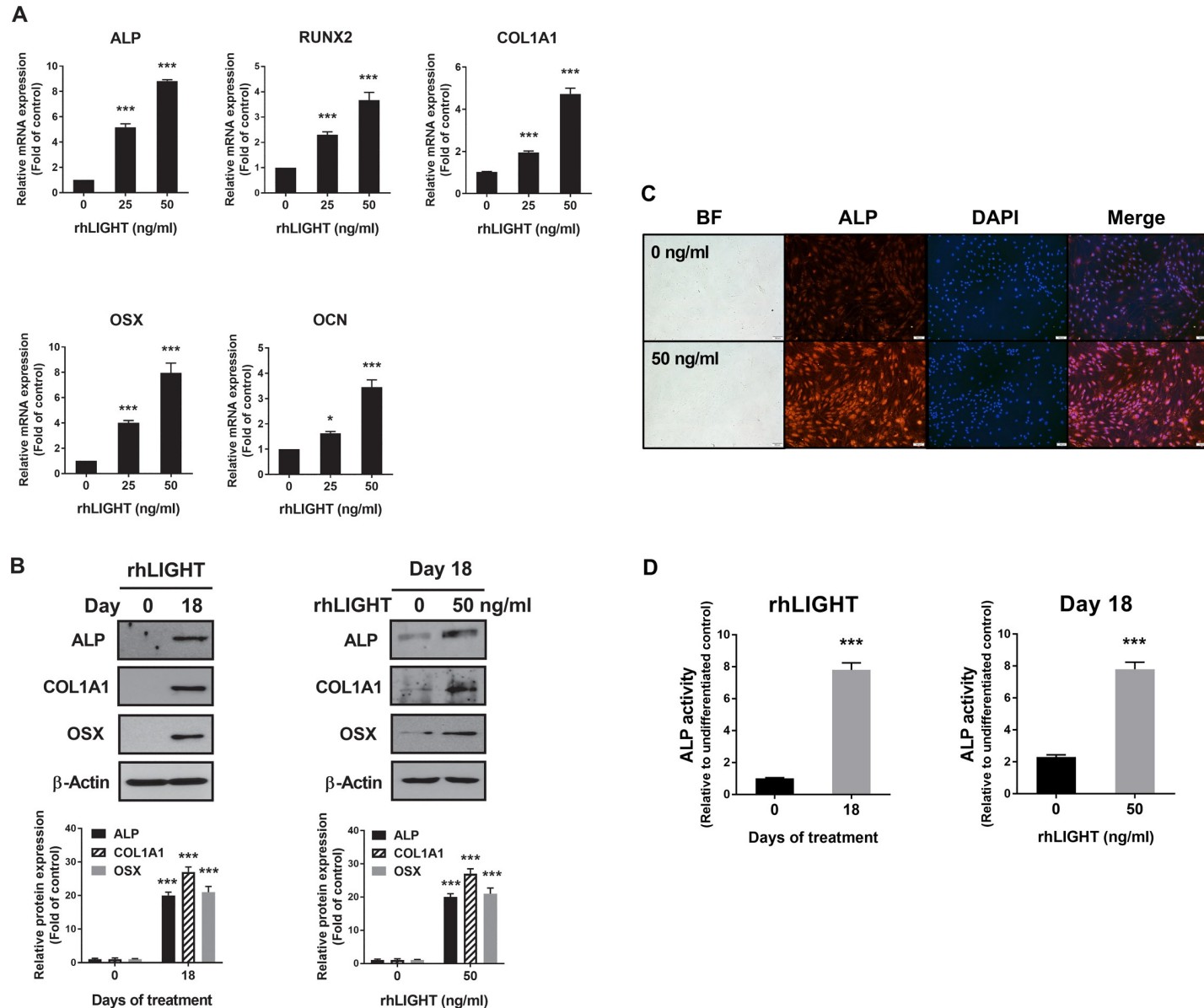

**Fig 3. RhLIGHT enhances the expression of osteoblast differentiation markers and alkaline phosphatase (ALP), and ALP activity in hBM-MSCs.** (A) Cells were stimulated with 0 and 50 ng/ml concentration of rhLIGHT for 18 days. The mRNA expressions of ALP, RUNX2, COL1A1, OSX, OCN by RT-qPCR increased in a dose-dependent manner. (B) Cells were stimulated with 0 and 50 ng/ml rhLIGHT for 18 days. The protein expression of ALP, COL1A1, and OSX by immunoblotting was increased time- and dose-dependently. (C) Cells were stimulated with 0 and 50 ng/ml concentrations of rhLIGHT for 18 days. The expression of ALP-positive cells (red) and nucleus (blue) was detected by fluorescence microscopy. Scale bar = 100 μm. (D) RhLIGHT enhances the ALP activity of hBM-MSCs time- and dose-dependently. Data represent the mean ± SEM. Significantly different from the control cells (*); *, $P < 0.05$; ***, $P < 0.001$. ALP, Alkaline phosphatase; RUNX2, Runt-related transcription factor 2; COL1A1, Collagen, type I, alpha 1; OSX, Osterix; OCN, Osteocalcin.

hBM-MSCs. Therefore, the expression level of LTβR increased depending on the concentration of rhLIGHT, as determined using RT-qPCR and western blotting analyses (Fig 5A and 5B, respectively).

## Discussion

Osteoporosis is a bone disease that increases the risk of fracture owing to decreased bone mass and density. In general, the goals of osteoporosis treatment are to reduce bone loss and

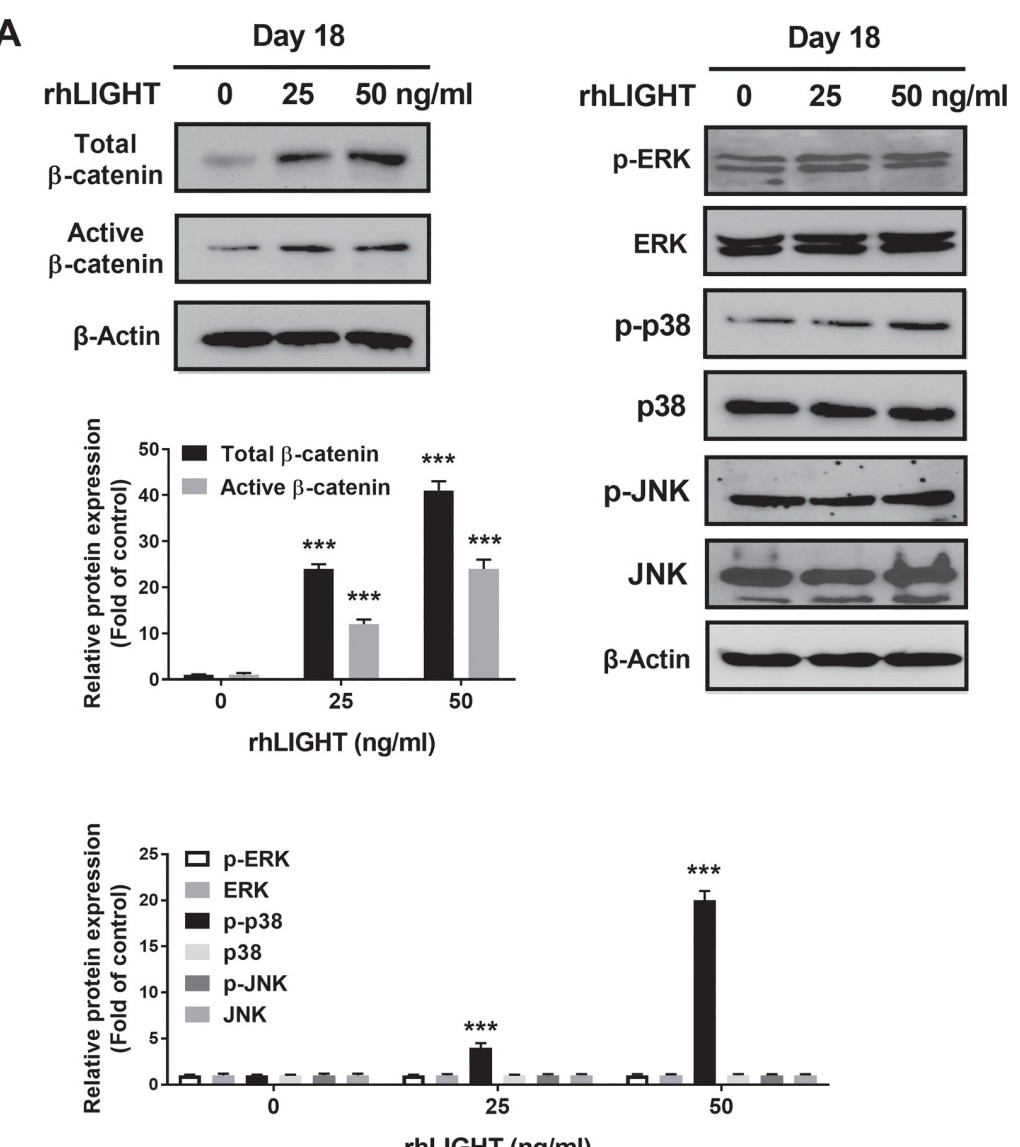

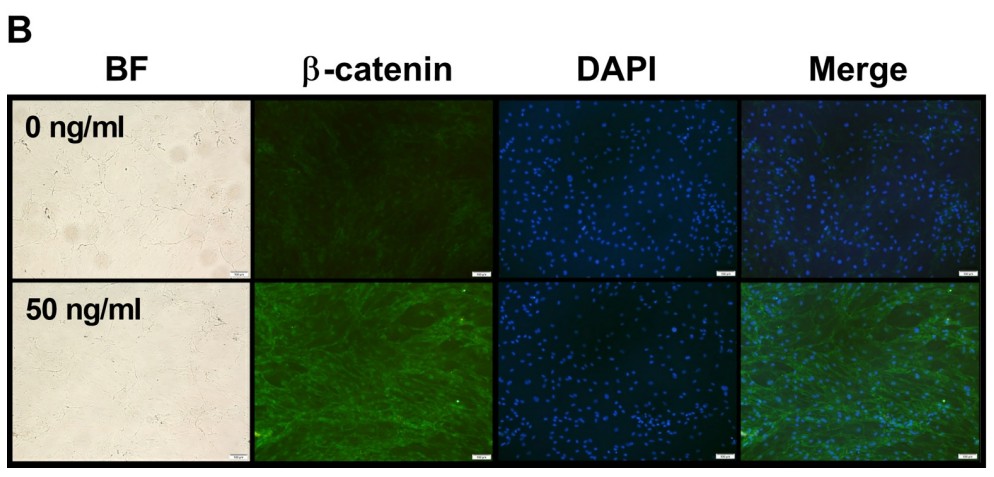

**Fig 4. RhLIGHT activates β-catenin in hBM-MSCs.** (A) Cells incubated with 0, 25, and 50 ng/ml concentrations of rhLIGHT for 18 days. The expression of total β-catenin, non-phospho (active) β-Catenin (Ser33/37/Thr41), p-ERK, ERK, p-p38, p38, p-JNK, and JNK was examined by immunoblotting. The membrane was stripped and reprobed with anti-β-actin mAb to confirm equal loading. (B) The expression of total β-catenin-positive cells (green) and nucleus (blue) by fluorescence microscopy. Scale bar: 100 μm.

maintain bone density, particularly in patients who have fractures or are at a high risk of fractures [1, 2]. The current treatment approach for osteoporosis typically consists of antiresorptive and osteoanabolic agents [4]. However, these drugs are associated with severe adverse effects. In particular, patients receiving long-term bisphosphonate treatment may exhibit osteonecrosis of the jaw and atypical femoral fractures. Thus, new drugs and treatments must be developed. Recent clinical studies have reported the development of various treatments for osteoporosis [28]. However, osteoporosis treatment remains poorly researched. In particular, cell replacement therapy via the use of MSCs may serve as a treatment strategy for osteoporosis in the future.

Stem cells have immense therapeutic potential, especially in the field of regenerative medicine. In particular, stem cells may be a promising source for cell-based treatments for osteoporosis. Therefore, cell replacement therapy using MSCs may be an option for osteoporosis treatment strategies in the future [2, 5]. Research on new treatments using stem cells derived from bone marrow from experimental animals or humans and various studies on regenerative or alternative cell-based treatments for osteoporosis are currently in progress. For example, in a rabbit model of osteoporosis, autologous BM-MSC transplantation was shown to improve bone formation [29], and in goats with long-term estrogen deficiency, which mimics postmenopausal osteoporosis in humans, such a treatment method improved bone formation and strengthened osteoporotic bones [30]. Similar results were obtained in a rat model of osteoporosis that was induced by ovariectomy after receiving allogeneic BM-MSCs isolated from healthy rats [31]. More recently, clinical trials have been using autologous BM-MSCs for osteoporosis treatment. In these studies, cells undergo fucosylation before intravenous infusion in patients with osteoporosis. However, the study is currently recruiting participants and is not yet complete (NCT ID: NCT02566655).

LTβR has various functions in regulating immune responses against pathogens, in organ development, and in maintaining structural integrity of secondary lymphoid tissues via its receptor LTα1β2 [32]. However, the use of LTβR for stem cell or bone formation and bone regeneration has been seldom reported. The interaction of LIGHT (other ligand of LTβR) and LTβR in hBM-MSCs is also poorly known.

Bone marrow is the most commonly used tissue source for adult MSCs [33]. BM-MSCs have been extensively studied for bone regeneration and repair because they are highly efficient in osteogenic differentiation [34]. In addition, cartilage regeneration based on BM-MSCs by chondrogenic preconditioning and mechanical stimulation has synergistic effects [35]. Thus, we selected BM-MSCs and conducted experiments related to bone formation. In a previous research, LTβR was constitutively expressed on the surface of hBM-MSCs (S1C Fig). LIGHT/LTβR interaction increased the number, viability, and proliferation of BM-MSCs, and it also promoted cell division by increasing the expression of CDK1 and CDK2 as well as various cyclins. In particular, it increased S/G2/M phase in the cells. Moreover, LIGHT-induced PDGF and TGFβ production mediated by STAT3 and Smad3 activation accelerated BM-MSC proliferation. Therefore, LIGHT/LTβR interaction increased the survival and proliferation of hBM-MSCs [10].

The present study revealed that the LIGHT/LTβR interaction is crucial in the osteogenic differentiation of hBM-MSCs. To the best of our knowledge, this study is the first to investigate

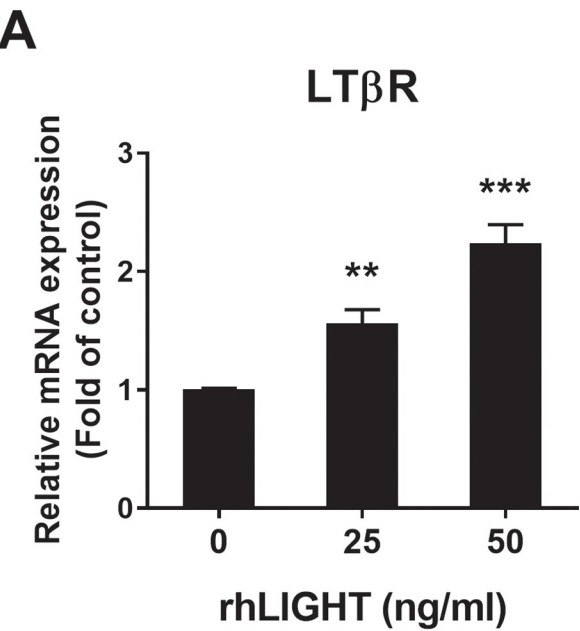

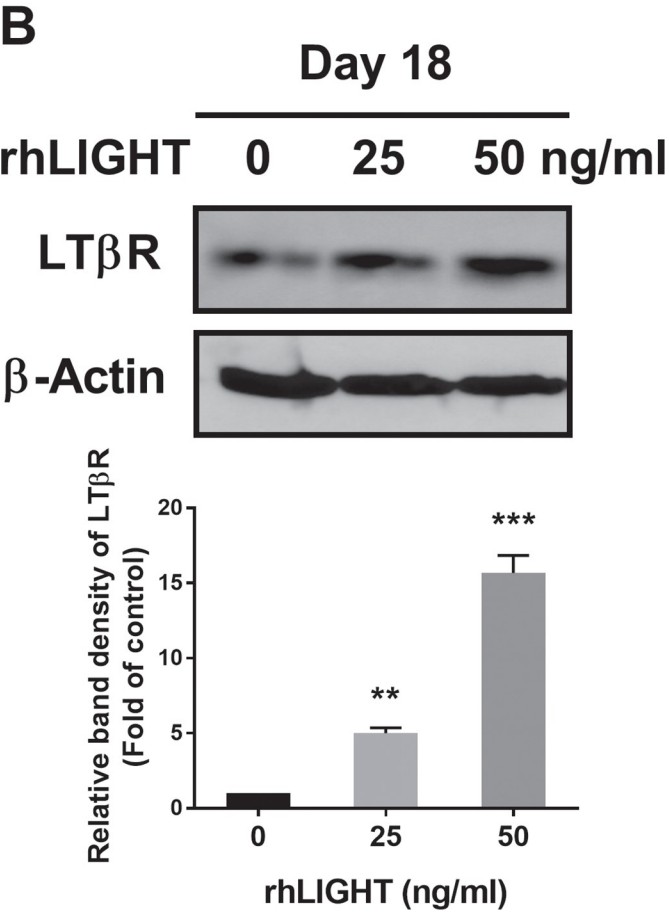

**Fig 5. RhLIGHT enhances the expression of the LIGHT receptor LTβR in hBM-MSCs.** (A) Cells were stimulated with 0 and 50 ng/ml concentration of rhLIGHT for 18 days. The mRNA expression of LTβR by RT-qPCR was increased dose-dependently. (B) Cells were stimulated with 0, 25, and 50 ng/ml concentration of rhLIGHT for 18 days. The protein expression of LTβR by immunoblotting was increased on day 18 dose-dependently. The membrane was stripped and reprobed with anti-β-actin mAb to confirm equal loading. Data represent the mean ± SEM. Significantly different from the control cells (*); **, $P < 0.01$; ***, $P < 0.001$. LTβR, Lymphotoxin beta receptor.

the effect of the LIGHT/LTβR interaction on the osteogenic differentiation of hBM-MSCs. In addition, their interaction increased calcium and phosphate deposition in hBM-MSCs (Figs 1 and 2). LIGHT/LTβR interaction also increased the expression of bone formation-related genes and proteins, such as ALP, RUNX2, COL1A1, OSX, and OCN, and strongly promoted the osteogenic differentiation of hBM-MSCs through the Wnt/β-catenin signaling pathway (Figs 3 and 4). The expression level of LTβR was increased by rhLIGHT treatment dose-dependently (Fig 5). In this study, we made great efforts to show that LIGHT (TNFSF14) enhances osteogenesis of hBM-MSCs through various experiments. In particular, in Fig 3, experiments were mainly conducted with only two concentrations. For western blotting analysis using hBM-MSCs, a very large amount of cells were required, and the incubation time after LIGHT treatment is too long. It was more than 18~21 days. The experiments were very difficult by all of these conditions. While conducting these studies more efficiently, the experiment was conducted by selecting two concentrations including the highest concentration to show dramatic changes in the results.

Our findings show that the LIGHT/LTβR interaction is critical for hBM-MSC osteogenesis. Therefore, LIGHT might play an important role in stem cell therapy for osteoporosis. In particular, LIGHT protein might be a good option to improve the quality of cell therapy that is limited by time. It usually takes at least 24–28 days for bone formation in hBM-MSCs, even with a medium for bone formation. However, shortening the time by approximately 7–10 days using rhLIGHT protein is an extremely important outcome that can take one step closer to the success of cell therapy application.

Transplantation therapy using MSCs may represent a clinically relevant solution for the treatment of osteoporosis, given their interesting properties and the promising results of pre-clinical and clinical studies. However, over the past few years, concerns have been raised about the long-term effectiveness of MSC-based therapy and the potential tumorigenic risk, there is a lack of standardized protocols for MSC transplantation. For all these reasons, we believe that further studies, especially randomized controlled trials, are needed to evaluate the long-term safety and efficacy of MSC-based treatments.

In conclusion, rhLIGHT-induced osteogenesis of hBM-MSCs was strongly mediated by WNT/β-catenin pathway activation, and it accelerated the differentiation of hBM-MSCs into osteocytes. In addition, LIGHT/LTβR interaction increases osteogenesis of hBM-MSCs. Therefore, LIGHT might play an important role in stem cell therapy.

## Supporting information

**S1 Fig. Quality test on hBM-MSCs.** (A) Negative marker (CD34, CD45, and CD19) staining in BM-MSCs. (B) Positive marker (CD90, CD44, and CD105) staining in BM-MSCs. (C) LTβR expression on the hBM-MSC surface. Each marker's expression level was determined using FACS analysis. Filled histogram represents the isotype control (mouse IgG); open histogram represents each antigen.
(DOCX)

**S1 Raw images.**
(PDF)

## Author Contributions

**Conceptualization:** Sook-Kyoung Heo, Eui-Kyu Noh, Jae-Cheol Jo.

**Data curation:** Sook-Kyoung Heo, Eui-Kyu Noh.

**Formal analysis:** Eui-Kyu Noh.

**Funding acquisition:** Sook-Kyoung Heo, Eui-Kyu Noh.

**Investigation:** Sook-Kyoung Heo, Yunsuk Choi, Yoo Kyung Jeong, Lan Jeong Ju, Ho-Min Yu, Do Kyoung Kim, Hye Jin Seo, Yoo Jin Lee, Jaekyung Cheon, SuJin Koh, Young Joo Min, Eui-Kyu Noh.

**Methodology:** Sook-Kyoung Heo, Yunsuk Choi, Lan Jeong Ju, Eui-Kyu Noh.

**Software:** Sook-Kyoung Heo, Yunsuk Choi, Eui-Kyu Noh.

**Supervision:** Sook-Kyoung Heo, Yunsuk Choi, Eui-Kyu Noh, Jae-Cheol Jo.

**Validation:** Sook-Kyoung Heo.

**Visualization:** Sook-Kyoung Heo, Eui-Kyu Noh.

**Writing – original draft:** Sook-Kyoung Heo.

**Writing – review & editing:** Sook-Kyoung Heo, Eui-Kyu Noh, Jae-Cheol Jo.

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
