## [Decision Letter · Decision Letter 0]

2 Dec 2020

PONE-D-20-35427

LIGHT (TNFSF14) enhances osteogenesis of human bone marrow-derived mesenchymal stem cells

PLOS ONE

Dear Dr. Jo,

Thank you for submitting your manuscript to PLOS ONE. After careful consideration, we feel that it has merit but does not fully meet PLOS ONE’s publication criteria as it currently stands. Therefore, we invite you to submit a revised version of the manuscript that addresses the points raised during the review process.

This study, albeit of interest needs to be consistently amended as specified in deep by the two referees. This because it is affected by flaws in all its parts.

The Authors must follow all the criticisms addressed by the referees and highlight all the amendments made in the text in submitting their revised version.

We look forward to receiving your revised manuscript.

Kind regards,

Gianpaolo Papaccio, M.D., Ph.D.

Academic Editor

PLOS ONE

Journal Requirements:

"This work was supported by the Basic Science Research Program through the National Research Foundation of Korea (NRF) funded by the Ministry of Education, Science and Technology (NRF-2018R1D1A1B07051343, NRF-2017R1A1A3A04069314). The research was also supported by the Basic Science Research Program through the Biomedical Research Center, funded by the Ulsan University Hospital (UUHBRC-2016-001). The funders had no role in study design, data collection and analysis, decision to publish, or preparation of the manuscript."

Reviewers' comments:

Reviewer's Responses to Questions

**Comments to the Author**

1. Is the manuscript technically sound, and do the data support the conclusions?

Reviewer #1: Partly

Reviewer #2: Yes

2. Has the statistical analysis been performed appropriately and rigorously? 

Reviewer #1: Yes

Reviewer #2: Yes

3. Have the authors made all data underlying the findings in their manuscript fully available?

Reviewer #1: Yes

Reviewer #2: Yes

4. Is the manuscript presented in an intelligible fashion and written in standard English?

Reviewer #1: No

Reviewer #2: No

5. Review Comments to the Author

Reviewer #1: In this study Authors evaluated the effects of LIGHT on osteogenesis of hBM-MSCs.

The study is interesting but some corrections need to be made.

English should be revised by a native speaker.

In introduction section, Authors talk about the different sources of stem cells: among them many studies have shown the ability of dental pulp cells to differentiate into different lineages (Cells. 2019 Mar 5; 8 (3): 217; Stem Cells Transl Med. 2020 Apr; 9 (4): 445-464).

In figure 1B Authors indicate the number of calcium deposits, but it would be more correct to quantize the staining with an appropriate protocol. Furthermore Authors should present a better 1A figure, particularly at 25 ng / ml the red background is too high.

Figure 2A should be lightened: it is too dark and this makes it impossible to visualize the result correctly.

Why did the authors only perform PCR up to 7 days and WB up to 18 days? It would be necessary to compare the data from the same days.

All IF images should be presented at higher quality: there is too much blue background.

Authors should specify why, after testing 2 concentrations, they choose to continue experiments with only the highest concentration.

Reviewer #2: In this manuscript, the authors aimed to evaluate the effect of CD258 on osteogenic differentiation of MSCs in order to treat osteoporosis. The study is of interest because it highlights the possibility to use new drugs for osteoporosis treatment. Although this, there are some points that need to be addressed.

In the Introduction section, the Authors should enlarge the part regarding MSCs with studies such as Cell Death and Disease, 2017, 8(1), e2568.

In figure 1, the Authors must add the scale bars. Moreover, they must quantify the Alizarin Red S and, in addition, they should analyze the ARS staining at different times starting from 7 days up to 21 days (7, 14 and 21days).

In figure 2A, the authors must change the pictures. It is difficult to detect the staining.

The Authors performed RT-qPCR and WB at different times (7 and 18 days, respectively), so there is no correlation between bone markers analyzed by RT-qPCR and those analyzed by western blot. First of all, RT-qPCR and Western blot must be performed at the same time and then, the molecular data must be confirmed by corresponding protein levels.

In figure 4, Western blot evaluation must be supplemented with corresponding densitometry.

In addition, the microarray analysis is not reported in M&M. Moreover, in figure 6, in order to show differentially expressed genes, the data must be reported as heat map analysis. Then, in the Result section, the Authors report some genes differentially expressed. These must be confirmed by real time-PCR and western blot. In my opinion, this part must be deleted or better evaluated with other experiments.

The Discussion must be revised. The first part describes results of other papers. The Authors must focus on their results, describe the novelty and limitations of work.

6. PLOS authors have the option to publish the peer review history of their article (what does this mean?). If published, this will include your full peer review and any attached files.

Reviewer #1: No

Reviewer #2: No

---

## [Author Response · Author response to Decision Letter 0]

27 Jan 2021

December 2nd, 2020

Gianpaolo Papaccio, M.D., Ph.D.

Academic Editor

PLOS ONE

Thank you very much for your letter dated December 2nd, 2020 concerning the status of our manuscript entitled, “LIGHT (TNFSF14) enhances osteogenesis of human bone marrow-derived mesenchymal stem cells” (Manuscript no, PONE-D-20-35427). The authors appreciate the constructive and helpful comments from the reviewers. We have revised the manuscript in accordance with the reviewers’ comments as detailed in the point-by point responses below.

Reviewers' comments:

Reviewer's Responses to Questions

Comments to the Author

1. Is the manuscript technically sound, and do the data support the conclusions?

Reviewer #1: Partly

Reviewer #2: Yes

2. Has the statistical analysis been performed appropriately and rigorously? 

Reviewer #1: Yes

Reviewer #2: Yes

3. Have the authors made all data underlying the findings in their manuscript fully available?

Reviewer #1: Yes

Reviewer #2: Yes

4. Is the manuscript presented in an intelligible fashion and written in standard English?

Reviewer #1: No

Reviewer #2: No

5. Review Comments to the Author

Reviewer #1

In this study Authors evaluated the effects of LIGHT on osteogenesis of hBM-MSCs.

The study is interesting but some corrections need to be made.

1. English should be revised by a native speaker.

Response: A comprehensive revision for language and mistypings were to be modified by a native speaker. In addition, we have attached an English proofreading certificate.

2. In introduction section, Authors talk about the different sources of stem cells: among them many studies have shown the ability of dental pulp cells to differentiate into different lineages (Cells. 2019 Mar 5; 8 (3): 217; Stem Cells Transl Med. 2020 Apr; 9 (4): 445-464).

Response: We agree with your comments. 

We have added the following words to the Introduction section (page 4, line 79): 

“dental pulp,”.

Also, we have added the following references to the References section (page 20, line 413): 

“Shi X, Mao J, Liu Y (2020) Pulp stem cells derived from human permanent and deciduous teeth: Biological characteristics and therapeutic applications. Stem Cells Transl Med 9: 445-464”.

3. In figure 1B Authors indicate the number of calcium deposits, but it would be more correct to quantize the staining with an appropriate protocol. Furthermore Authors should present a better 1A figure, particularly at 25 ng/ml the red background is too high.

Response: We agree with your comments. 

We conducted an experiment again, and detected the new images of calcium deposition in presence of rhLIGHT protein at different times starting from 7 days up to 21 days, as shown in as shown in Supplementary Fig. 1A and 1B.

Deposition of calcium was assessed using Alizarin red staining from 7 days up to 21 days after osteogenic induction by rhLIGHT, as shown in as shown in Fig. 1. In the samples on day 14 and day 18, calcium deposition was observed only in the rhLIGHT treatment group. Moreover, in the sample on day 21, calcium deposition began to be observed in the control group, and the degree of calcium deposition was accelerated in the rhLIGHT treatment group. Relative intensity of phosphate deposition by Alizarin red staining in Figure 1A is summarized by Figure 1B. According to Fig. 1A and 1B, rhLIGHT effect was better on day 18 than day 21. Accordingly, we used these conditions for the remainder of the experiments.

Supplementary Fig. 1. RhLIGHT increases calcium deposition in hBM-MSCs.

(A) Cells were incubated with 0, 25, and 50 ng/ml concentrations of rhLIGHT for 7, 14, 18 and 21 days and then stained with 2% Alizarin red to confirm calcium deposits, as described in the Materials and Methods. Microscope magnification: ×100, Scale bar = 100 μm. (B) The number of calcium deposition spots in Figure 1A is summarized by Figure 1B. Data are expressed as mean ± SEM. Significantly different from the control cells (*); ***, P < 0.001.

Thus, these new observations are presented as the Figure 1 in the revised manuscript.

So, we have added new Figure 1A and Figure 1B in the revised manuscript.

We have added the following sentences to the Results section (page 12, line 224):

“Deposition of calcium was assessed using Alizarin red staining from 7 days up to 21 days after osteogenic induction by rhLIGHT, as shown in as shown in Fig. 1. In the samples on day 14 and day 18, calcium deposition was observed only in the rhLIGHT treatment group. Moreover, in the sample on day 21, calcium deposition began to be observed in the control group, and the degree of calcium deposition was accelerated in the rhLIGHT treatment group. Relative intensity of phosphate deposition by Alizarin red staining in Figure 1A is summarized by Figure 1B. According to Fig. 1, rhLIGHT effect was better on day 18 than day 21. Accordingly, we used these conditions for the remainder of the experiments”.

We have added the following words to the Figure legends section (page 24, line 486):

“Fig. 1. RhLIGHT increases calcium deposition in hBM-MSCs.

(A) Cells were incubated with 0, 25, and 50 ng/ml concentrations of rhLIGHT for 7, 14, 18 and 21 days and then stained with 2% Alizarin red to confirm calcium deposits, as described in the Materials and Methods. Microscope magnification: ×100, Scale bar = 100 μm. (B) Relative intensity of phosphate deposition by Alizarin red staining in Figure 1A is summarized by Figure 1B. Data are expressed as mean ± SEM. Significantly different from the control cells (*); ***, P < 0.001.”

4. Figure 2A should be lightened: it is too dark and this makes it impossible to visualize the result correctly.

Response: We agree with your comments. 

We have fixed them.

Thus, these new pictures are presented as the Figure 2A in the revised manuscript.

So, we have added new Figure 2A in the revised manuscript.

5. Why did the authors only perform PCR up to 7 days and WB up to 18 days? 

It would be necessary to compare the data from the same days.

Response: We agree with your comments. 

We conducted an experiment again, and detected mRNA by using RT-qPCR analyses in presence of rhLIGHT protein at several concentrations (0, 25, and 50 ng/ml) on 18 days, as shown in as shown in Supplementary Fig. 2.

Supplementary Fig. 2. Cells were stimulated with 0 and 50 ng/ml concentration of rhLIGHT for 18 days. The mRNA expressions of ALP, RUNX2, COL1A1, OSX, OCN by RT-qPCR increased in a dose-dependent manner. *, P < 0.05; ***, P < 0.001. ALP, Alkaline phosphatase; RUNX2, Runt-related transcription factor 2; COL1A1, Collagen, type I, alpha 1; OSX, Osterix; OCN, Osteocalcin.

Thus, these new observations are presented as the Figure 3A in the revised manuscript.

So, we have added new Figure 3A in the revised manuscript.

We have added the following sentences to the Results section (page 13, line 241):

“RT-qPCR analysis revealed that the mRNA levels of ALP, RUNX2, COL1A1, OSX, and OCN on day 18 were significantly high following treatment with 50 ng/ml rhLIGHT treatment (Fig. 3A)”.

We have added the following words to the Figure legends section (page 24, line 502): 

“(A) Cells were stimulated with 0 and 50 ng/ml concentration of rhLIGHT for 18 days. The mRNA expressions of ALP, RUNX2, COL1A1, OSX, OCN by RT-qPCR increased in a dose-dependent manner”.

Supplementary Fig. 3. RhLIGHT enhances the expression of the LIGHT receptor LTβR in hBM-MSCs. Cells were stimulated with 0 and 50 ng/ml concentration of rhLIGHT for 18 days. The mRNA expression of LTβR by RT-qPCR was increased dose-dependently. Data represent the mean ± SEM. Significantly different from the control cells (*); **, P < 0.01; ***, P <0.001. LTβR, Lymphotoxin beta receptor.

Thus, these new observations are presented as the Figure 5A in the revised manuscript.

So, we have added new Figure 5A in the revised manuscript.

We have added the following sentences to the Results section (page 14, line 281):

“Therefore, the expression level of LTβR increased depending on the concentration of rhLIGHT, as determined using RT-qPCR and western blotting analyses (Fig. 5A and 5B, respectively)”.

We have added the following words to the Figure legends section (page 25, line 521): 

“(A) Cells were stimulated with 0 and 50 ng/ml concentration of rhLIGHT for 18 days. The mRNA expression of LTβR by RT-qPCR was increased dose-dependently”.

6. All IF images should be presented at higher quality: there is too much blue background.

Response: We agree with your comments. We have fixed them.

Thus, these new pictures are presented as the Figure 3C and Figure 4B in the revised manuscript.

So, we have added new Figure 3C and Figure 4B in the revised manuscript.

7. Authors should specify why, after testing 2 concentrations, they choose to continue experiments with only the highest concentration.

Response: We agree with your comments. 

In this study, we made great efforts to show that LIGHT (TNFSF14) enhances osteogenesis of human bone marrow-derived mesenchymal stem cells through various experiments. In particular, in Fig. 3, experiments were mainly conducted with only two concentrations. For western blotting analysis using human bone marrow-derived mesenchymal stem cells, a very large amount of cells were required, and the incubation time after LIGHT treatment is too long. It was more than 18~21 days. The experiments were very difficult by all of these conditions. While conducting these studies more efficiently, the experiment was conducted by selecting two concentrations including the highest concentration to show dramatic changes in the results.

We have added the following sentences to the Discussion section (page 16, line 333):

“In this study, we made great efforts to show that LIGHT (TNFSF14) enhances osteogenesis of hBM-MSCs through various experiments. In particular, in Fig. 3, experiments were mainly conducted with only two concentrations. For western blotting analysis using hBM-MSCs, a very large amount of cells were required, and the incubation time after LIGHT treatment is too long. It was more than 18~21 days. The experiments were very difficult by all of these conditions. While conducting these studies more efficiently, the experiment was conducted by selecting two concentrations including the highest concentration to show dramatic changes in the results”.

Finally, the authors appreciate the constructive and helpful comments from the reviewers.

Thank you so much.

 

Reviewer #2

In this manuscript, the authors aimed to evaluate the effect of CD258 on osteogenic differentiation of MSCs in order to treat osteoporosis. The study is of interest because it highlights the possibility to use new drugs for osteoporosis treatment. Although this, there are some points that need to be addressed.

1. In the Introduction section, the Authors should enlarge the part regarding MSCs with studies such as Cell Death and Disease, 2017, 8(1), e2568.

Response: We agree with your comments. And we understand your points.

Generally, cancer cells and stem cells share several biological properties, and transcriptional factors that monitor the fate of stem cells could play a major role in the renewal of cells modified from the cancerous microenvironment. Therefore, the induction and maintenance of a stemness phenotype in mesenchymal cells might be a further mechanism of survival and resistance to drugs implemented by the tumor. The identification of factors secreted by cancer cells that induce such changes or specific markers that characterize the new modified cells could be useful in strengthening the conventional treatments and combat the relapse of the disease. Consequently, adipose grafts may give rise, in the case of cancer cell persistence after surgery to tumour growth. Therefore, it must be strongly discouraged in groups of patients including those undergoing:adipose graft after a breast cancer for mastoplasty; adipose graft, following cancer in general for every treatment. In these circumstances, the use of adipose tissue for auto-grafting must be carefully performed, only after meticulous analyses of possible cancer.

We have added the following sentences to the Introduction section (page 5, line 90):

“However, there are concerns about the therapeutic approach using MSCs. Because cancer cells and stem cells share several biological properties, transcription factors that monitor the fate of stem cells may play an important role in the regeneration of modified cells in the microenvironment of cancer”. 

We have added the following sentences to the Discussion section (page 17, line 347):

“Transplantation therapy using MSCs may represent a clinically relevant solution for the treatment of osteoporosis, given their interesting properties and the promising results of preclinical and clinical studies. However, over the past few years, concerns have been raised about the long-term effectiveness of MSC-based therapy and the potential tumorigenic risk, there is a lack of standardized protocols for MSC transplantation. For all these reasons, we believe that further studies, especially randomized controlled trials, are needed to evaluate the long-term safety and efficacy of MSC-based treatments”.

2. In figure 1, the Authors must add the scale bars. Moreover, they must quantify the Alizarin Red S and, in addition, they should analyze the ARS staining at different times starting from 7 days up to 21 days (7, 14 and 21days).

Response: We agree with your comments. 

We have fixed them. First of all, we have added the scale bars.

As the reviewer commented, we conducted an experiment again, and detected the new images of calcium deposition in presence of rhLIGHT protein at different times starting from 7 days up to 21 days, as shown in as shown in Supplementary Fig. 1A and 1B.

Deposition of calcium was assessed using Alizarin red staining from 7 days up to 21 days after osteogenic induction by rhLIGHT, as shown in as shown in Fig. 1. In the samples on day 14 and day 18, calcium deposition was observed only in the rhLIGHT treatment group. Moreover, in the sample on day 21, calcium deposition began to be observed in the control group, and the degree of calcium deposition was accelerated in the rhLIGHT treatment group. Relative intensity of phosphate deposition by Alizarin red staining in Figure 1A is summarized by Figure 1B. According to Fig. 1A and 1B, rhLIGHT effect was better on day 18 than day 21. Accordingly, we used these conditions for the remainder of the experiments.

Supplementary Fig. 1. RhLIGHT increases calcium deposition in hBM-MSCs.

(A) Cells were incubated with 0, 25, and 50 ng/ml concentrations of rhLIGHT for 7, 14, 18 and 21 days and then stained with 2% Alizarin red to confirm calcium deposits, as described in the Materials and Methods. Microscope magnification: ×100, Scale bar = 100 μm. (B) The number of calcium deposition spots in Figure 1A is summarized by Figure 1B. Data are expressed as mean ± SEM. Significantly different from the control cells (*); ***, P < 0.001.

Thus, these new observations are presented as the Figure 1 in the revised manuscript.

So, we have added new Figure 1A and Figure 1B in the revised manuscript.

We have added the following sentences to the Results section (page 12, line 224):

“Deposition of calcium was assessed using Alizarin red staining from 7 days up to 21 days after osteogenic induction by rhLIGHT, as shown in as shown in Fig. 1. In the samples on day 14 and day 18, calcium deposition was observed only in the rhLIGHT treatment group. Moreover, in the sample on day 21, calcium deposition began to be observed in the control group, and the degree of calcium deposition was accelerated in the rhLIGHT treatment group. Relative intensity of phosphate deposition by Alizarin red staining in Figure 1A is summarized by Figure 1B. According to Fig. 1, rhLIGHT effect was better on day 18 than day 21. Accordingly, we used these conditions for the remainder of the experiments”.

We have added the following words to the Figure legends section (page 24, line 486):

“Fig. 1. RhLIGHT increases calcium deposition in hBM-MSCs.

(A) Cells were incubated with 0, 25, and 50 ng/ml concentrations of rhLIGHT for 7, 14, 18 and 21 days and then stained with 2% Alizarin red to confirm calcium deposits, as described in the Materials and Methods. Microscope magnification: ×100, Scale bar = 100 μm. (B) Relative intensity of phosphate deposition by Alizarin red staining in Figure 1A is summarized by Figure 1B. Data are expressed as mean ± SEM. Significantly different from the control cells (*); ***, P < 0.001.”

3. In figure 2A, the authors must change the pictures. It is difficult to detect the staining.

Response: We agree with your comments. 

We have fixed them. 

Thus, these new pictures are presented as the Figure 2A in the revised manuscript.

So, we have added new Figure 2A in the revised manuscript.

4. The Authors performed RT-qPCR and WB at different times (7 and 18 days, respectively), so there is no correlation between bone markers analyzed by RT-qPCR and those analyzed by western blot. First of all, RT-qPCR and Western blot must be performed at the same time and then, the molecular data must be confirmed by corresponding protein levels. 

Response: We agree with your comments. 

We conducted an experiment again, and detected mRNA by using RT-qPCR analyses in presence of rhLIGHT protein at several concentrations (0, 25, and 50 ng/ml) on 18 days, as shown in as shown in Supplementary Fig. 2.

Supplementary Fig. 2. Cells were stimulated with 0 and 50 ng/ml concentration of rhLIGHT for 18 days. The mRNA expressions of ALP, RUNX2, COL1A1, OSX, OCN by RT-qPCR increased in a dose-dependent manner. *, P < 0.05; ***, P < 0.001. ALP, Alkaline phosphatase; RUNX2, Runt-related transcription factor 2; COL1A1, Collagen, type I, alpha 1; OSX, Osterix; OCN, Osteocalcin.

Thus, these new observations are presented as the Figure 3A in the revised manuscript.

So, we have added new Figure 3A in the revised manuscript.

We have added the following sentences to the Results section (page 13, line 241):

“RT-qPCR analysis revealed that the mRNA levels of ALP, RUNX2, COL1A1, OSX, and OCN on day 18 were significantly high following treatment with 50 ng/ml rhLIGHT treatment (Fig. 3A)”.

We have added the following words to the Figure legends section (page 24, line 502): 

“(A) Cells were stimulated with 0 and 50 ng/ml concentration of rhLIGHT for 18 days. The mRNA expressions of ALP, RUNX2, COL1A1, OSX, OCN by RT-qPCR increased in a dose-dependent manner”.

Supplementary Fig. 3. RhLIGHT enhances the expression of the LIGHT receptor LTβR in hBM-MSCs. Cells were stimulated with 0 and 50 ng/ml concentration of rhLIGHT for 18 days. The mRNA expression of LTβR by RT-qPCR was increased dose-dependently. Data represent the mean ± SEM. Significantly different from the control cells (*); **, P < 0.01; ***, P <0.001. LTβR, Lymphotoxin beta receptor.

Thus, these new observations are presented as the Figure 5A in the revised manuscript.

So, we have added new Figure 5A in the revised manuscript.

We have added the following sentences to the Results section (page 14, line 281):

“Therefore, the expression level of LTβR increased depending on the concentration of rhLIGHT, as determined using RT-qPCR and western blotting analyses (Fig. 5A and 5B, respectively)”.

We have added the following words to the Figure legends section (page 25, line 521): 

“(A) Cells were stimulated with 0 and 50 ng/ml concentration of rhLIGHT for 18 days. The mRNA expression of LTβR by RT-qPCR was increased dose-dependently”.

5. In figure 4, Western blot evaluation must be supplemented with corresponding densitometry.

Response: We agree with your comments.

We agree with your comments that the quantification of band intensity based on beta actin by densitometry analysis for the western blot results, is also desirable and would greatly improve the presentation of results.

Therefore, we have added the relative band density (target protein/β-actin) in new Figure 3B and Figure 4A in the revised manuscript.

6. In addition, the microarray analysis is not reported in M&M. Moreover, in figure 6, in order to show differentially expressed genes, the data must be reported as heat map analysis. Then, in the Result section, the Authors report some genes differentially expressed. These must be confirmed by real time-PCR and western blot. In my opinion, this part must be deleted or better evaluated with other experiments.

Response: We agree with your comments. 

And we have removed the all results and statements related to microarray analysis in the revised manuscript.

7. The Discussion must be revised. The first part describes results of other papers. 

The Authors must focus on their results, describe the novelty and limitations of work.

Response: Response: We agree with your comments. 

And we have removed the all results related to other papers in the first part of discussion. Also, we have revised the many part in the Discussion.

In this study, we made great efforts to show that LIGHT (TNFSF14) enhances osteogenesis of human bone marrow-derived mesenchymal stem cells through various experiments. For western blotting analysis using human bone marrow-derived mesenchymal stem cells, a very large amount of cells were required, and the incubation time after LIGHT treatment is too long. It was more than 18~21 days. The experiments were very difficult by all of these conditions”.

We have added the following sentences to the Discussion section (page 16, line 333):

“In this study, we made great efforts to show that LIGHT (TNFSF14) enhances osteogenesis of hBM-MSCs through various experiments. 

We have added the following sentences to the Discussion section (page 17, line 336):

“For western blotting analysis using hBM-MSCs, a very large amount of cells were required, and the incubation time after LIGHT treatment is too long. It was more than 18~21 days. The experiments were very difficult by all of these conditions”.

We have added the following sentences to the Discussion section (page 17, line 347):

“Transplantation therapy using MSCs may represent a clinically relevant solution for the treatment of osteoporosis, given their interesting properties and the promising results of preclinical and clinical studies. However, over the past few years, concerns have been raised about the long-term effectiveness of MSC-based therapy and the potential tumorigenic risk, there is a lack of standardized protocols for MSC transplantation. For all these reasons, we believe that further studies, especially randomized controlled trials, are needed to evaluate the long-term safety and efficacy of MSC-based treatments”.

Finally, the authors appreciate the constructive and helpful comments from the reviewers.

Thank you so much.

Once again, we thank the editors and reviewers for their careful evaluation of our work and we hope that our manuscript is now considered suitable for publication in PLOS ONE

Sincerely yours,

Jae-Cheol Jo, M.D., Ph.D.

Department of Hematology and Oncology, 

Ulsan University Hospital, 

University of Ulsan College of Medicine, 

877 Bangeojinsunhwan-doro, Dong-gu, Ulsan 44033, Republic of Korea. 

Tel: +82-52-250-8632

Fax: +82-52-252-7048

E-mail address: jcjo97@hanmail.net

6. PLOS authors have the option to publish the peer review history of their article (what does this mean?). If published, this will include your full peer review and any attached files.

Do you want your identity to be public for this peer review? For information about this choice, including consent withdrawal, please see our Privacy Policy.

Reviewer #1: No

Reviewer #2: No

---

## [Decision Letter · Decision Letter 1]

8 Feb 2021

LIGHT (TNFSF14) enhances osteogenesis of human bone marrow-derived mesenchymal stem cells

PONE-D-20-35427R1

Dear Dr. Jo,

We’re pleased to inform you that your manuscript has been judged scientifically suitable for publication and will be formally accepted for publication once it meets all outstanding technical requirements.

Kind regards,

Gianpaolo Papaccio, M.D., Ph.D.

Academic Editor

PLOS ONE

Additional Editor Comments (optional):

Reviewers' comments:

Reviewer's Responses to Questions

**Comments to the Author**

1. If the authors have adequately addressed your comments raised in a previous round of review and you feel that this manuscript is now acceptable for publication, you may indicate that here to bypass the “Comments to the Author” section, enter your conflict of interest statement in the “Confidential to Editor” section, and submit your "Accept" recommendation.

Reviewer #1: All comments have been addressed

Reviewer #2: All comments have been addressed

2. Is the manuscript technically sound, and do the data support the conclusions?

Reviewer #1: (No Response)

Reviewer #2: Yes

3. Has the statistical analysis been performed appropriately and rigorously? 

Reviewer #1: (No Response)

Reviewer #2: Yes

4. Have the authors made all data underlying the findings in their manuscript fully available?

Reviewer #1: (No Response)

Reviewer #2: Yes

5. Is the manuscript presented in an intelligible fashion and written in standard English?

Reviewer #1: (No Response)

Reviewer #2: Yes

6. Review Comments to the Author

Reviewer #1: (No Response)

Reviewer #2: (No Response)

7. PLOS authors have the option to publish the peer review history of their article (what does this mean?). If published, this will include your full peer review and any attached files.

Reviewer #1: No

Reviewer #2: No

---

## [Editor Report · Acceptance letter]

10 Feb 2021

PONE-D-20-35427R1 

LIGHT (TNFSF14) enhances osteogenesis of human bone marrow-derived mesenchymal stem cells 

Dear Dr. Jo:

I'm pleased to inform you that your manuscript has been deemed suitable for publication in PLOS ONE. Congratulations! Your manuscript is now with our production department. 

Kind regards, 

on behalf of

Prof. Gianpaolo Papaccio 

Academic Editor

PLOS ONE